# Ecological Trait Differences Are Associated with Gene Expression in the Primary Visual Cortex of Primates

**DOI:** 10.3390/genes16020117

**Published:** 2025-01-22

**Authors:** Trisha M. Zintel, John J. Ely, Mary Ann Raghanti, William D. Hopkins, Patrick R. Hof, Chet C. Sherwood, Jason M. Kamilar, Amy L. Bauernfeind, Courtney C. Babbitt

**Affiliations:** 1Department of Biology, University of Massachusetts Amherst, Amherst, MA 01003, USA; trisha.m.zintel@gmail.com; 2MAEBIOS, Alamogordo, NM 88310, USA; john10ely@gmail.com; 3Department of Anthropology, Kent State University, Kent, OH 44242, USA; mraghant@kent.edu; 4Keeling Center for Comparative Medicine and Research, The University of Texas, MD Anderson Cancer Center, Bastrop, TX 78602, USA; wdhopkins@mdanderson.org; 5Nash Family Department of Neuroscience and Friedman Brain Institute, Icahn School of Medicine at Mount Sinai, New York, NY 10029, USA; patrick.hof@mssm.edu; 6New York Consortium in Evolutionary Primatology, New York, NY 10065, USA; 7Department of Anthropology and Center for the Advanced Study of Human Paleobiology, The George Washington University, Washington, DC 20052, USA; sherwood@email.gwu.edu; 8Department of Anthropology, University of Massachusetts Amherst, Amherst, MA 01003, USA; jkamilar@umass.edu; 9Organismic and Evolutionary Biology Graduate Program, University of Massachusetts Amherst, Amherst, MA 01003, USA; 10Department of Neuroscience, Washington University School of Medicine in St. Louis, St. Louis, MO 63110, USA; amybauernfeind@wustl.edu; 11Department of Anthropology, Washington University in St. Louis, St. Louis, MO 63110, USA

**Keywords:** brain, evolution, metabolic, phenotype, genomics

## Abstract

Primate species differ drastically from most other mammals in how they visually perceive their environments, which is particularly important for foraging, predator avoidance, and detection of social cues. Background/Objectives: Although it is well established that primates display diversity in color vision and various ecological specializations, it is not understood how visual system characteristics and ecological adaptations may be associated with gene expression levels within the primary visual cortex (V1). Methods: We performed RNA-Seq on V1 tissue samples from 28 individuals, representing 13 species of primates, including hominoids, cercopithecoids, and platyrrhines. We explored trait-dependent differential expression (DE) by contrasting species with differing visual system phenotypes and ecological traits. Results: Between 4–25% of genes were determined to be differentially expressed in primates that varied in type of color vision (trichromatic or polymorphic di/trichromatic), habitat use (arboreal or terrestrial), group size (large or small), and primary diet (frugivorous, folivorous, or omnivorous). Conclusions: Interestingly, our DE analyses revealed that humans and chimpanzees showed the most marked differences between any two species, even though they are only separated by 6–8 million years of independent evolution. These results show a combination of species-specific and trait-dependent differences in the evolution of gene expression in the primate visual cortex.

## 1. Introduction

In most mammals, vision is critically important for the purposes of foraging, predator avoidance, and mate recognition [1,2,3,4,5]. However, primates are distinguished from other mammals by unique visual traits, including high visual acuity and variation in color perception [2,5,6,7,8,9]. Primates with forward-facing eyes have high orbital convergence and correspondingly enhanced binocular vision, resulting in greater depth perception (stereoscopic vision) and increased visual acuity [10]. Primates also possess a uniquely specialized fovea, a localization of cone receptors in the retina, that allows for greater resolution [11,12]. Among primates, visual acuity is highest in diurnal haplorrhine species [13,14,15,16,17]. Trichromatic color vision is a hallmark of hominoid and cercopithecoid species, but color vision varies significantly across platyrrhine species, as well as within some species that have polymorphic sex-linked di/trichromacy [18].

Higher visual acuity and a shift to trichromatic color vision are hypothesized to confer several benefits across primate species. High visual acuity has been suggested to be adaptively favorable in relation to diet (i.e., detecting insects), as well as for arboreal species navigating tree limbs [10]. At the level of opsin gene conservation across primates and tree shrews, the typical light conditions that require enhanced visual acuity are the same as those in which color vision is favored, representing a putative adaptive link between enhanced acuity and trichromacy [8]. The evolution of a third opsin gene increases the range of detectable wavelengths (especially in the range of red hues) and has been hypothesized to be influenced by foraging and social pressures [19,20]. The foraging hypothesis stresses the importance of detecting red ripe fruits and young leaves against a green foliage background [1,2,3,21,22]. The social signal hypothesis states that wavelength sensitivity for red hues is important for the accurate interpretation of skin color changes indicative of receptivity, sociality, or health [19,23,24]. Evidence in a lemur species with mixed populations of trichromats and dichromats showed that there is a group-level benefit when at least a single trichromatic individual is present, likely linked to the ability to locate ripe fruit during the dry season, thus positively influencing fitness [22].

The primary visual cortex (V1) is the first cortical region in the visual pathway and where visual inputs converge from both eyes. Inputs from both eyes that impart binocular vision are critical for detecting edge orientation, thus playing an important role in depth perception [9,25]. Increased orbital convergence allowing for binocular vision correlates with an increase in size in brain regions associated with visual perception [26]. The selectivity of neurons in the V1 for orientation, as well as sensitivity to signal disparities between retinal signals, are important for depth perception [12,27,28,29]. Neuronal density within the V1 of primates is more than twice that of other mammals, reflecting the importance of vision for primates [9,10]. There is also a known variation in V1 neuronal cytoarchitecture across hominoid species, including molecular differences in cytochrome oxidase staining in layer 4A [30,31] and interneuron density [32,33] as measured by immunohistochemistry. Taken together, these findings may indicate an adaptive advantage of visual processes that could differentially influence V1 function in primates. However, we still have little knowledge about the genetic mechanisms driving V1 variation across species and how this correlates with differences in visual acuity and perception.

King and Wilson [34] hypothesized that the extensive phenotypic differences between humans and chimpanzees must be due to changes in gene expression rather than coding sequence evolution, given the high degree of similarity at the nucleotide and protein levels. Investigations of differential expression (DE) across primates have demonstrated that *cis*-regulatory changes are critical for large-scale differences in phenotype [35,36,37,38,39]. Gene expression tends to vary in different tissues of the body, and different regions of the brain are known to have region-specific profiles indicative of localized regulation [40,41,42,43,44,45]. Enrichments for neuron-specific processes (e.g., synaptic signaling) and metabolism (e.g., carbohydrate and lipid metabolism) have been reported in human brain gene expression in comparison to that of the chimpanzee and rhesus macaque [36,37,38] and may be indicative of region-specific differences in neural function.

There have been few investigations of interspecific gene expression differences of the visual cortex of primates [41,46,47,48]. Khaitovich et al. [41] investigated a number of brain regions, including the V1, with the primary focus on determining the extent of region-specific differences in expression between species. These authors found that regions of the cerebral cortex, including the V1, differ significantly from other brain regions, such as the cerebellum, and that the degree of intraspecific variation in expression between these brain regions was significantly greater in humans than chimpanzees. Furthermore, they reported that in the cerebral cortex of both species, the V1 was an outlier among the other regions investigated (dorsolateral prefrontal cortex, anterior cingulate cortex, and Broca’s area [41]. Another study found co-expression networks specific to the V1 in both humans and chimpanzees and found the greatest degree of interspecific conservation in the V1 compared to any other region investigated [48]. The authors attributed these findings to the sensory nature of the V1’s function, for which humans and chimpanzees presumably differ very little [48]. A comparison of transcriptions between human, macaque, and mouse V1s using microarrays found that primate (human and macaque) V1 expression was conserved in comparison to mouse, suggesting that the conserved genomic profiles of the V1 extends beyond primates [46]. These studies all represent important initial investigations of V1 gene expression; however, they largely investigate only a few primate species (humans, chimpanzees, rhesus macaques) and emphasize other brain regions over the V1.

To better understand the molecular basis of visual system adaptations, we used RNA-Seq to quantify gene expression in the V1 of 13 primate species across three major clades (hominoids, cercopithecoids, and platyrrhines), significantly increasing the number of primates for which V1 gene expression data exist. This broad phylogenetic sampling of phenotypically diverse primates allowed us to explore how differences in gene expression were associated with variations in traits that contribute to a species’ visual perception, including differences in species-typical color vision (trichromats vs polymorphic di/trichromats) and ecological variables such as habitat use (arboreal vs terrestrial), average group size (large vs small), and typical diet (folivore vs frugivore vs omnivore). We also evaluated the influence of sex on gene expression in the V1. We observed statistically significant differences in V1 gene expression associated with variation in the visually relevant traits of color vision, habitat use, group size and diet, but not sex. These expression differences, as well as expression differences between species, are largely driven by altered metabolic signaling. Human and chimpanzee V1 expression is notably different from other species, and though they do not differ in the visual ecological traits investigated here, they are divergent from one another in important neurological signaling processes.

## 2. Materials and Methods

Samples and Library Preparation. Tissue samples were collected from 28 individuals representing 13 primate species (n = 1–3 per species; Figure 1, Appendix A). Frozen brain samples from captive adult primates free of known neurological disorders were obtained from various research institutions and zoos. All individuals had been cared for according to Federal and Institutional Animal Care and Use guidelines and died of natural causes. All tissue was collected and stored at −80 °C until RNA collection, with postmortem intervals of less than 8 h. Further details about the samples can be found in Appendix A. Due to the opportunistic nature of sampling such a phylogenetically broad group of primates, there is an unavoidable variation in the age and sex of some individuals sampled (Appendix A).

Each sample was dissected from the cortex of the medial aspect of the occipital pole, surrounding the calcarine sulcus. The thin, striate cortex of the V1 was visually identified in each sample. All dissections included the gray matter of the V1, extending a small extent into the underlying white matter. Tissue samples were homogenized using a TissueLyser (Qiagen, Hilden, Germany), prior to total RNA extraction using an RNeasy Plus Mini Kit (Qiagen), including a DNase step to remove residual DNA. Total RNA was analyzed for quality using the Agilent Bioanalyzer system (Agilent RNA 6000 Nano Kit, Santa Clara, CA, USA). RNA integrity varied among our samples due to sampling from deceased primates, but there was no bias for species. All RINs were above 7, and there was no bias in the biological replicates across individuals within a species. Using the NEBNext Poly(A) Magnetic mRNA Isolation Kit (NEB, Ipswich, MA, USA), mRNA was isolated from intact total RNA, and cDNA libraries were made from each sample using the NEBNext RNA Library Prep Kit for Illumina (NEB). The library quality was assessed using the Agilent DNA 1000 Kit. Pooled samples were sequenced using the Illumina NextSeq 500 platform at the UMass Amherst (Amherst, MA, USA) genomics core to produce 75 base pair reads, yielding a minimum of 20 million reads per sample. All the read data, in the form of FASTQ files, have been submitted to the National Center for Biotechnology Information (NCBI)’s Short Read Archive (SRA), with accession number PRJNA526359.

Read Mapping and Quantification. Quality-filtered reads were aligned to available primate genomes with Bowtie2 using default parameters for gapped alignments [49]. Current species-specific ENSEMBL genomes were used (*Homo sapiens*, GRCh38.p10; *Pan troglodytes*, CHIMP2.1.4; *Gorilla gorilla*, gorGor3.1; *Pongo abellii*, PPYG2; *Papio anubis*, PapAnu2.0; *Nomascus leucogenys*, Nleu1.0; *Macaca mulatta*, Mmul_8.0.1; *Callithrix jacchus*, C_jacchus3.2.1) [50,51]. For species for which there is no publicly available reference genome (*Pongo pygmaeus*, *Symphalangus syndactylus*, *Macaca nemestrina*, *Erythrocebus patas*, *Saimiri sciureus*, *Pithecia pithecia*, and *Ateles fusciceps*), reads were mapped to the closest related primate for which there was a genome available. Specifically, the *M. nemestrina* and *E. patas* reads were mapped to *M. mulatta*, *S. syndactylus* to *N. leucogenys*, *A. fusciceps*, *S. sciureus*, and *P. pithecia* to *C. jacchus*, and *P. pygmaeus* to *P. abellii*. Mapping percentages were ≥80% using default “—local” Bowtie2 parameters. The “—very-sensitive-local” parameter was used to increase the accuracy and alignment percentage of the samples with the lowest mapping percentages (<than 85%; 2 *A. fusciceps*, 1 *S. syndactylus*, and 1 *S. sciureus* sample), and all increased to ≥83.5% (see Appendix A). HT-Seq [52] was used to quantify counts per gene for each sample, using ENSEMBL gene transfer files (GTFs) corresponding to the same genome build used for alignment [53]. For each species, homologous genes were matched to the ENSEMBL human reference set of genes using biomaRt [54]. These were subsequently filtered for the 12,564 genes from all species that have a high orthology confidence, all of which also had a high homolog percent identity to the human query genes as well as gene order conservation scores, as previously determined by ENSEMBL [53]. This resulted in the removal of genes from all the transcriptomes considered if they did not meet orthology confidence in all species. Finally, we used the R package edgeR4.4.1 [55] to filter out lowly expressed genes (counts per million (CPM) > 1 in 1/28 samples), resulting in 12,330 expressed orthologs in our dataset.

Clustering Analyses. We used clustering analyses to determine the variation in our samples. We produced a principal coordinates analysis (PCoA) using the expression profiles of all of the protein coding genes for each of our samples (n = 1–3 per species). To further visualize patterns within our data, we produced phenograms by performing a hierarchical clustering of the PCoA distances. The PCoA and dendrograms show that our samples largely cluster by species and clade.

Differential Expression Analysis. Differential expression (DE) analyses were performed for multiple pairwise comparisons using a generalized linear model (GLM) in the R package edgeR (Appendix A) [55]. Gene expression was considered significantly different at a false discovery rate (FDR) of less than 5%. The DE between species only included species for which we had more than one sample, including human, chimpanzee, siamang, olive baboon, rhesus macaque, pig-tailed macaque, patas monkey, spider monkey, and marmoset (Appendix A), to incorporate intraspecific variability into the calculation of statistical significance. To determine if there was a relationship between the number of genes exhibiting DE between species, we used a Pearson correlation between the numbers of genes exhibiting DE and the divergence times reported by 10kTrees of the last common branching point of the tree between the two species compared (Appendix A [56]. The multi-factor GLM allowed us to analyze all the data at once to detect DE between species, as well as between groups of samples of which the species are known to have distinct differences in visually relevant phenotypes, data for which were collected from the literature [4,10,57,58,59,60]. These species-typical data were used to label each of our samples for color visual system (trichromats or polymorphic tri/dichromats), average group size (less than 20 individuals per group or greater than 20 individuals per group), primary diet (frugivorous, folivorous, or omnivorous), and primary habitat (arboreal or terrestrial) (mapped onto a phylogeny in Figure 1; listed in Appendix A). For each of these phenotype contrasts, all samples of the same species-typical phenotype were grouped together and compared to the group of samples with the opposing species-typical phenotype (e.g., all samples from primarily frugivorous species were grouped together and compared to all samples from primarily omnivorous species for the phenotype–DE comparison, frugivorous versus omnivorous diet). DE was also performed on each sample by sex (male or female) (Appendix A). Given the comparative, discovery-based nature of our analyses, we did not correct the *p*-values of DE genes for the multiple interspecies or phenotype–DE comparisons, though many would have remained significant. Our goal was to investigate differences in V1 gene expression between species, but also to attempt to link V1 expression differences with differences in broad, species-level phenotypes. It is important to note that there a few caveats to this approach relevant to the interpretation of the results. We used species-typical categories for each of the phenotypes we were interested in correlating with V1 gene expression, because we do not have the relevant phenotypic information on each of the individuals from which our brain samples were obtained. We also only had tissue from captive animals. For these reasons, we labeled our samples for correlations with phenotype using broad categories (e.g., primarily arboreal species vs primarily terrestrial species rather than more discrete measurements), and we determined DE by grouping samples by species-typical phenotype and comparing between phenotypically distinct groups. This leads to another important caveat for our phenotype–DE comparisons, which was that our analyses did not allow us to account for phylogeny. We “mapped” these traits onto a phylogeny in Figure 1 to provide transparency about some of the clear influences of phylogeny on our divergent phenotype groups. For example, the investigated hominoid and cercopithecoid species are monomorphic for trichromatic color vision in all individuals regardless of genotype or sex, while the platyrrhine species are polymorphic, with trichromatic homozygous females, dichromatic males, or dichromatic–heterozygous females (Figure 1; Appendix A). With these caveats in mind, we were conservative in our interpretation of these analyses.

**Figure 1 genes-16-00117-f001:**
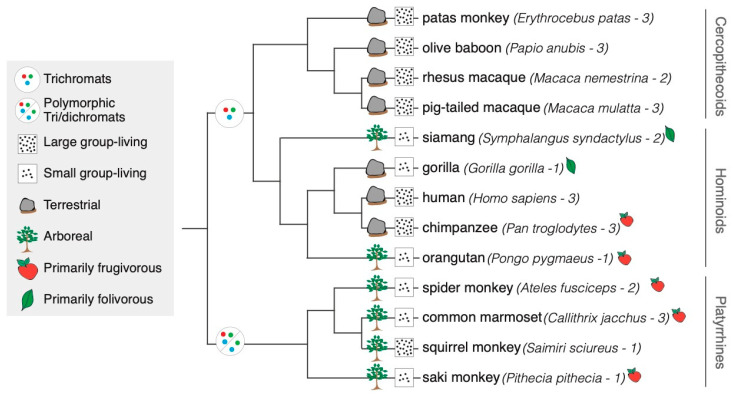
Phenotypic traits of species for which V1 gene expression was investigated. Phenotypic traits for color vision, habitat use, and group size are mapped on the phylogeny, and diet is depicted to the right of the species names. The number of individuals is next to the species names. Species without a diet indicator were coded as omnivorous. The tree was generated using 10kTrees Version 3 [56] and Mesquite [61].

Categorical Enrichment Analysis of Differentially Expressed Genes. Genes identified as showing DE for each interspecies and phenotype comparison were subsequently used for pathway enrichment with the Kyoto Encyclopedia of Genes and Genomes (KEGG) [62] to determine differences in signaling, as well as Gene Ontology (GO) cellular component (CC) categorical enrichment analyses [63,64] to further determine the cellular locality of differences. Enrichments were obtained using EnrichR with *crisp* datasets and filtered for those that had an enrichment *p*-value less than 0.05 [65,66]. To determine the most biologically relevant enrichments, the KEGG pathway enrichments were further filtered to include only those with five or more genes per enrichment, and the GO CC enrichments were further filtered to only include those with ten or more genes per enrichment. Due to the discovery-based nature of our research question and the filtering of categories based upon gene number, we did not restrict our analyses to only categories with significant adjusted *p*-values for multiple comparisons, of which there were fewer. To obtain an overall view of the global processes enriched in primate V1 DE, we grouped specific pathways into broad parent KEGG categories for “signaling”, “immune”, “disease”, “neuron-specific”, “metabolic”, and “other” (Appendix A), generally following the established hierarchy already present in the KEGG ontology. Similarly, GO CC enrichment terms were also grouped into parent categories related to cellular structures: e.g., “neural cell projection” is a larger category containing GO CC terms such as “axon” and “dendritic branch point”, which are encompassed in the established GO CC hierarchy of “cell projections” (Appendix A). The parent category of KEGG “neuron-specific processes” includes enrichments for pathways that are unique to the brain, and the majority of these were for synaptic processes (specific neurotransmitters, synaptic activity, and synaptic vesicle function; Appendix A). Metabolic pathways include those related to the metabolic breakdown or synthesis of all major macromolecules, including that of carbohydrates (e.g., “glycolysis/gluconeogenesis”), lipids (e.g., “fatty acid metabolism”), amino acids (e.g., “glycine, serine, and threonine metabolism”), and nucleotides (e.g., “purine metabolism”; Appendix A). We report all significant categorical enrichments (Appendix A), but in our results, we largely focus on enrichments for pathways specifically involved in neuronal and metabolic signaling, as previous work has found that these processes exhibit DE and signatures of positive selection in *cis*-regulatory regions when comparing human, chimpanzee, and rhesus macaque brains [36,37,38] and the importance of metabolism in the brain to provide cellular energy and the critical synthesis and breakdown of macromolecules [67,68,69,70,71].

## 3. Results

Humans and chimpanzees exhibit the most variation in V1 gene expression.

Generally, V1 gene expression profiles were grouped by species and phylogenetic clade (Figure 2). There was a greater variation in V1 transcriptomes within hominoids than cercopithecoids or platyrrhines, predominantly due to the distances observed among samples representing human and chimpanzee (Figure 2A). Thus, the expression profiles of other hominoids (orangutan, gorilla, and siamang) were more similar to that of cercopithecoids and platyrrhines than to human and chimpanzee (Figure 2A). Despite the fact that gene expression does not evolve in the same manner as nucleotide sequences [72], due to its tissue and cell-type specificity within an organism, the hierarchical clustering of whole transcriptomes revealed similar clustering results, with most samples per species clustering generally according to phylogeny (Figure 2B). The exceptions were for the single representative samples of gorilla, saki monkey, squirrel monkey, and orangutan. Additionally, the two pig-tailed macaque samples clustered away from the other cercopithecoid species. These multivariate analyses indicate conservation in the overall expression profiles of primates except for human and chimpanzee. There are no overt technical factors influencing the out-grouping of human, chimpanzee, or pig-tailed macaque samples (e.g., individual sex or age, sample hemisphere of origin, RNA or cDNA library quality, read number, alignment percentages, Appendix A). Our finding of human and chimpanzee divergence from other species is consistent with other studies [35,38,44,45] and highlights that the uniqueness in brain gene expression with proximity to the human lineage extends to the V1. Next, we performed pairwise DE analyses between species (Appendix A). There was a positive correlation between the number of genes exhibiting DE and phylogenetic distance (r = 0.6, *p* = 0.0001, Pearson correlation), demonstrating that the number of genes exhibiting DE increases with phylogenetic distance (Appendix A). However, humans exhibit noticeably greater numbers of genes displaying DE compared to the relatively closely related species chimpanzee (n = 3648) and siamang (n = 3846) (Appendix A). Notably, chimpanzees and siamangs do not exhibit as high a degree of difference from one another (n = 2771) (Appendix A). This may suggest that there is a significant difference in V1 expression with phylogenetic proximity to humans and not more generally in the hominoid clade. These results demonstrate that there is a significant degree of DE in the V1 across primate species, despite its sensory function.

Metabolic and neuronal signaling are enriched in V1 DE.

To determine what biological processes are enriched in genes displaying DE in the V1, we performed categorical enrichment analyses for KEGG pathway terms of pairwise DE comparisons (Table 1; Appendix A). Overall, there was a wide variety of KEGG pathways enriched in these DE comparisons related to the inter- and extracellular signaling processes associated with growth and development, as well as more specific pathways related to metabolism (e.g., glycolysis) and neuron-specific signaling (e.g., synaptic signaling) (Appendix A; for more details on the thresholding and grouping of pathways into parent categories, see Section 2). To assess whether metabolic and neural processes routinely differ in the gene expression of primate V1s, as well as what interspecies DE comparisons deviated from that trend, we determined what proportion of all the pathways enriched were metabolic or neuron-specific signaling pathways. On average, metabolic pathways accounted for 23.8% of enrichments, while neural pathways only accounted for 7.6% of enrichments (Appendix A). Similarly, enrichments for metabolism were present in all interspecies DE comparisons (34/34 pairwise species comparisons), while enrichments for neuron-specific processes were present in only 23/34 comparisons (Appendix A). Interestingly, there was a reduction in the proportion of metabolic enrichments observed in the human–chimpanzee DE comparison, coinciding with an increase in the number of neuron-specific enrichments increasing (Table 1; Appendix A).

The synaptic signaling pathways included in the neuron-specific parent group of enriched pathways included glutamatergic, cholinergic, serotonergic, and GABAergic synaptic signaling, as well as “synaptic vesicle cycling” and “long-term potentiation” (Appendix A). Other neuron-specific KEGG pathways that were enriched included axon guidance, neurotrophin signaling, and oxytocin signaling (Appendix A). All eight DE comparisons of chimpanzees to other species had higher proportions of neuron-specific pathway enrichments of any interspecies DE comparison (Appendix A). The DE comparison between human and chimpanzee had the most neuron-specific enrichments of any interspecies DE comparison (n = 12; Appendix A), including the cholinergic and GABAergic synapse among the top ten most significantly enriched pathways (*p* < 0.0001; Table 1). In contrast, there were no neuron-specific pathways enriched in the top ten KEGG pathways for any other human interspecies DE comparisons (except neuroactive ligand receptor interaction in the DE comparison between human and siamang) (Table 1). These results highlight that differences in neuron-specific processes in the V1 are not common across all primates but are more prominent with proximity to humans, most notably between human and chimpanzee.

The V1 region is highly specialized in cytoarchitecture and structure [9], and previous studies have determined that many of the V1-specific differences in gene expression between species are for genes involved in determining this structure [46]. Therefore, in addition to KEGG pathway enrichments, we conducted categorical enrichment analyses for the GO CC for pairwise DE comparisons, to determine if there were differences in primate V1s related to specific neuron cellular parts. In many interspecific comparisons, including but not limited to human vs. chimpanzee, there were several GO CC enrichments related to plasma membrane-associated complexes of known importance for intercellular signaling in the brain, including ATP-coupled ion channels and various neurotransmitter plasma membrane receptors and the intermembrane transporters the “clathrin-sculpted glutamate transport vesicle membrane”, “ionotropic glutamate receptor complex”, and “ciliary neurotrophic factor receptor complex” (Appendix A). There were also enrichments for GO CC terms for specialized cell projections that characterize neural cells, including components of dendrites (e.g., dendritic branch points) and types of axons (“C fiber”), as well as for synapse parts and neurofilaments (Appendix A). Interestingly, DE comparisons between human and chimpanzee and other species are the only interspecies DE comparisons to be enriched for dendrite cellular parts, and only chimpanzee DE comparisons are enriched for axon cellular parts (Appendix A). The enrichments for GO cellular components corroborate the KEGG pathway enrichments and further show that neural signaling complexes critical for neurological signaling and structure, such as plasma membrane-associated channels, dendrites, and axons, differ among primate V1s.

To determine how metabolic gene expression differed from global gene expression in primate V1s, we subset our expression matrix to include only genes involved in metabolic KEGG pathways (n = 1039, Figure 3). When we generated a heatmap where species clustered based upon the distance between expression values, we saw that the expression profiles of metabolic genes result in the clustering of species according to clades: all platyrrhines, hominoids, and cercopithecoids group together, with no overt pattern in the metabolic KEGG pathways to which the genes belong (bottom color-coded bar, Figure 3). This demonstrates that while there are significant differences in metabolic gene expression in primate V1s, species-specific metabolic patterns still seem to contain substantial phylogenetic signals. To try to elucidate what metabolic processes distinguish each clade, we identified which pathways demonstrated DE between species of different clades but not between species belonging to the same clade. Carbohydrate metabolism appears to be the most distinguishing set of metabolic pathways between clades. Hominoids differ from platyrrhines in citrate (TCA) cycle gene expression and differ from both cercopithecoids and platyrrhines for glyoxylate and decarboxylate metabolism (Appendix A). In contrast, platyrrhines differ from both cercopithecoids and hominoids for amino sugar metabolism and glycolysis/gluconeogenesis gene expression (Appendix A). One noticeable exception of a carbohydrate metabolic pathway that differed consistently between species comparisons of all three cross-clade combinations but never between two species of the same clade was fructose and mannose metabolism, which displayed DE between thirteen interspecies comparisons, including six comparisons of cercopithecoids to platyrrhines, five comparisons of hominoids to cercopithecoids, and two comparisons of hominoids to platyrrhines (Appendix A). Taken together, these results show that metabolic changes in primate V1s are common across primate species in a manner consistent with phylogeny and that the metabolic gene expression differences influencing this are primarily related to carbohydrate metabolism. In summary, variation in primate V1 gene expression is not limited to neural signaling processes and in fact seems to be driven to a large extent by altered metabolic signaling, although neural signaling appears to contribute more to differences in expression with proximity to humans.

Visual cortex DE is correlated with differences in color vision, group size, diet, and habitat use.

We leveraged the wealth of phenotypic and behavioral data available for the sampled primate species to help understand correlations between V1 gene expression and variation in species-typical color vision, habitat use, diet, group size, and individual differences in sex (summarized in Figure 1; details in Section 2 and Appendix A) [4,10,57,58,59,60]. These phenotype–DE comparisons were conducted by grouping all the samples according to the species-typical state of their color visual system (trichromats or polymorphic di/trichromats), habitat use (primarily arboreal or terrestrial), generalized diet (primarily frugivorous, folivorous, or omnivorous), and group size (primarily living in small or large groups) and determining genes exhibiting DE between the phenotype extremes. Of these trait-based DE comparisons, we detected DE based upon differences in color visual system (trichromats–polymorphic di/trichromats), primary habitat use (arboreal–terrestrial), group living size (large–small), and primary diet (frugivore–folivore, folivore–omnivore, frugivore–omnivore) but not sex in the samples used for this study (Table 2, Appendix A). The highest proportion of trait-based DE was for differences in color vision (25.3%), and the lowest was for the diet-based comparisons (4.3–7.9%; Table 2, Appendix A). The grouping of such disparate species likely introduces far more variability (“noise”) than when comparing two discrete species and results in far fewer numbers of genes exhibiting DE in these phenotype comparisons. We reasoned that this would be a fairly conservative estimate of differential expression. While many of the parent categories of pathways were enriched in all phenotype–DE comparisons, there were unique enrichments for specific pathways depending on the phenotype compared (Table 3). It is important to note that for some traits investigated here, there is a clear influence of phylogeny on differences in phenotype based purely on the species for which we have samples (further discussed in Section 2). Because we cannot parse out how much this influences the phenotype–DE comparisons, we are conservative in our interpretations of these analyses.

Color vision has long been hypothesized to be adaptive in primates, with numerous complementary and competing hypotheses about the pressures influencing its evolution (reviewed in [18]). When comparing expression between species that differ in color vision, 25.3% (n = 3173) of genes display DE (Table 2; Appendix A). For our samples, the investigated hominoid and cercopithecoid species are monomorphic trichromats in all individuals, while the platyrrhine species are polymorphic with trichromatic homozygous females, dichromatic males, or dichromatic–heterozygous females (Figure 1; Appendix A). However, despite color vision being sexually dimorphic in some of the platyrrhine species, very few genes were significantly differentially expressed in V1s between sexes (0.28% DE, n = 35) (Table 2; Appendix A). There were no neuronal-specific KEGG signaling pathways enriched in genes displaying DE between trichromatic species and polymorphic di/trichromatic species, but there were a number of metabolic pathways enriched, including amino acid, nucleotide, glycolysis/gluconeogenesis, and sphingolipid metabolic pathways (Appendix A). Notably, color vision was the only phenotype–DE comparison enriched for the encompassing large KEGG pathways of “metabolic pathways” and “carbon metabolism”, and it included 226 genes (Appendix A). This is 8-fold more genes than any other KEGG pathway enrichment, suggesting that there are extensive metabolic differences in the V1s of primates differing in color vision. However, given the clear phylogenetic influence on these samples (Figure 1, Appendix A), we cannot parse out if these differences are truly due to divergent color visual perception or rather how hominoids and cercopithecoids differ from platyrrhines in the V1.

We investigated DE between primarily arboreal and terrestrial species due to the role of the V1 in processing signals derived from navigating and perceiving the environment [9]. The primarily arboreal species in our study included all four platyrrhine species and two hominoid species (orangutan and siamang), while the primarily terrestrial species included all four cercopithecoid species and three hominoid species (human, chimpanzee, and gorilla) (Figure 1, Appendix A). There were 2501 genes with DE in the V1 associated with differences in primate habitat use (either arboreal or terrestrial) (19.9%, Table 2; Appendix A). Neural KEGG pathways enriched in species differing in habit use included neuroactive ligand receptor interaction, long-term depression, and retrograde endocannabinoid signaling (Appendix A). This phenotype–DE comparison was enriched primarily for lipid metabolic pathways (Appendix A). Lipids are important in the brain for long-term energy storage, membrane structure, extracellular signaling, and the enhanced propagation of neural signaling (e.g., myelination) [73]. Habitat-use DE enrichments for lipid metabolism may suggest that these processes differ significantly in the V1 as it responds to visual stimuli from divergent interactions with habitat, such as the enhanced V1 processing of depth perception for arboreal versus terrestrial species.

We also explored possible expression differences in the V1 that correlate with differences in sociality, using group size as a proxy variable. For the included species, the “small group” included three platyrrhines species (spider monkey, marmoset, and saki monkey), and three hominoid species (siamang, gorilla, and orangutan), with fewer than 20 individuals per group on average, and the “large group” included all four cercopithecoid species, one platyrrhine species (squirrel monkey), and two hominoid species (human and chimpanzee), with greater than 20 individuals per group on average (Figure 1, Appendix A). We found that 13.6% (n = 1704) genes exhibited DE between species that differed in group size (Table 2; Appendix A). Categorical enrichment determined the genes displaying DE between species differing in group size had more neural processes enriched than any other DE comparison based upon trait except for diet (see below), including neuroactive ligand receptor interaction, long-term depression, retrograde endocannabinoid signaling, and the GABAergic synapse (Appendix A). Of the genes exhibiting DE between large and small group-living species, metabolic KEGG enrichments included amino acid sugar and nucleotide sugar metabolism and fatty acid degradation (Appendix A). Because both long-term depression and GABAergic synaptic signaling function in inhibitory signaling [74], enrichments for both of these processes may suggest a significant difference in inhibitory signaling in the V1 among primates differing in group size, which may be due in part to the differential visual responses necessary for navigating socially complex environments.

To compare V1 expression between species that differ by diet, we categorized our species as primarily frugivorous, primarily folivorous, or omnivorous [60]. The primarily frugivorous species included two hominoid species (chimpanzee and orangutan) and three platyrrhine species (spider monkey, marmoset, and saki monkey), while we had two primarily folivorous hominoid species (gorilla and siamang) (Figure 1, Appendix A). The omnivorous species included all four cercopithecoid species, a single platyrrhine species (squirrel monkey), and a single hominoid species (human) (Figure 1, Appendix A). We calculated DE among the three possible comparisons: folivore–frugivore (4.3% DE, n = 534), folivore–omnivore (5.3%, n = 664), and frugivore–omnivore (7.9%, n = 991) (Table 2; Appendix A). Folivorous species differed from omnivorous species in tryptophan, purine, and steroid metabolic KEGG pathways (Appendix A). Folivorous species differed from frugivorous species in carbohydrate and protein digestion and retinol and tryptophan metabolism (Table 3). Frugivorous primates displayed much higher numbers of genes exhibiting DE in the V1 than frugivore–folivore and folivore–omnivore comparisons, though this may be due to a lack of folivorous species in the available samples (only the siamang and gorilla were folivorous; Figure 1; Appendix A). Frugivorous–omnivorous DE was enriched for serotonergic synaptic signaling and was the only diet comparison enriched for any neuron-specific KEGG pathways (Appendix A). Frugivores differed from omnivores metabolically in terms of amino acid metabolism (Table 3). Given that serotonin levels have an established link to diet [75] and tryptophan is a known precursor for serotonin [76], the finding that both tryptophan and serotonergic signaling are only enriched in DE between species differing in diet may suggest that diet has an impact on V1 serotonin signaling by way of altered tryptophan metabolism.

All trait DE comparisons were enriched for at least one pathway in the metabolic parent categories but not for neural signaling (Appendix A). Our results suggest that neural processes drive differences in V1 gene expression in species that differ in group size, habitat use, and diet, while metabolic differences are more responsible for V1 DE between species differing in color vision (Appendix A). Our approach correlating DE with phenotype suggests that there is a link between expression in the V1 and distinctive differences in visually relevant traits.

## 4. Discussion

Although some studies of V1 gene expression have compared it to other regions of the brain, as well as across species [41,46,47,48], no other investigations of brain gene expression included nearly the diversity and number of primate species as in the current study. As such, this represents one of the first in-depth investigations of how V1 gene expression differs across a variety of primate species. We found that global gene expression in primate V1s clusters largely by phylogenetic relatedness. Carbohydrate metabolic processes seem to be driving expression differences across the primate tree, while neural processes are more conserved in the V1. A deviation from this trend is the expression differences between human and chimpanzee. These species are outliers from all other species and display a relatively large amount of intraspecific variation in global gene expression profiles, as well as in specific pathways related to synaptic signaling and neuronal cell projections important for maintaining the complex neurological signaling networks key to brain function. In addition to global and interspecific differences in expression, we were able to correlate V1 DE to differences in visually relevant phenotypes.

The enrichment for neuron-specific processes in V1 DE, primarily between human and chimpanzee, is consistent with previous findings of enrichments for similar processes, such as synaptic signaling, in the differential gene expression that has been reported in various studies of human, chimpanzee, and rhesus macaque brains [36,37,38]. Also, similar metabolic processes as those enriched here have also been found to differentiate primate brain gene expression in a number of previous studies of a smaller array of primate species [40,41,46,77], including that humans specifically differ largely in metabolic processes related to aerobic glycolysis [37,38]. Taken together, our results across a broad number of primate species’ V1s show that these trends for metabolic and neural signaling differences in the brain extend to the V1.

Given that the V1 is a primary sensory cortex and in view of subtle interspecies differences in visual perception between closely related primates, there might not be much divergence expected over shorter evolutionary times. Consistent with this, we found a significant correlation for greater numbers of genes exhibiting DE between more distantly related species. Our finding that interspecies DE tends to increase with evolutionary distance has been observed previously in a study of gene expression across ten species (including six hominoids) and six tissue types [42]. Furthermore, we observed a trend for genes involved in metabolic signaling to be differentially expressed consistently across interspecies DE comparisons, regardless of phylogenetic distance, while processes specific to neural signaling were far less common. Because the most metabolically demanding feature of the brain is synaptic transmission, metabolic differences are not mutually exclusive with neural signaling processes, but this is nonetheless an interesting trend. Our findings suggest that the altered gene expression of neural-specific pathways in the V1 does not consistently contribute to functional differences between closely related species, but altered metabolic processes do (e.g., oxidative phosphorylation is enriched in genes exhibiting DE between all cercopithecoid species; Appendix A). The exception to this is that neuron-specific processes were among the most significantly enriched pathways in genes exhibiting DE between human and chimpanzee.

Within primates, selective differences in the genome can be linked to diet and metabolism, suggesting selection has optimized different metabolic processes in lineage-dependent ways [36,37,38,78,79,80,81]. The human brain is more energetically costly than that of other primates, utilizing ~20% of all of the body’s metabolic resources, in comparison to non-human primate brains that use less than 10% [82,83]. Importantly, allometry alone does not explain the increase in human brain appropriation of glucose metabolism at this proportion [84,85,86]. Our results are consistent with previous findings that humans differ in brain gene expression from chimpanzees for neuron-specific processes related to synaptic transmission and metabolic processes involved in aerobic glycolysis [37,38,40,41,46,77]. These data demonstrate that, like other brain regions [35,38,44,45], human lineage-specific neurological changes are present in the visual cortex.

It is important to again note that there are a number of caveats to this approach that are relevant to the interpretation of the results. Importantly, our samples are from captive living animals that did not necessarily experience the wild ecological conditions typical for their species. Thus, our study design allows us to interpret minimal innate differences in V1 gene expression that may be related to past evolutionary influences from visual ecological pressures but not from dynamic experience-dependent differences, which would probably yield even stronger signals of ecological correlations. We also only have multiple individuals from some of the species represented here (Appendix A). This is not uncommon with primate species, where only opportunistic sampling is possible, but it is a limitation of our analyses.

Understanding V1 gene expression differences based upon a variation in phenotypes relevant for vision is an area of research that could elucidate the neurological implications of differences in vision and the selective pressures hypothesized to be linked to these traits. Although there are important caveats to our analysis of phenotype-based DE, primarily an inability to account for phylogenetic influences (see Section 2 for more details), this still represents a significant effort to link proximate gene expression differences in the brain with evolved variation in ecological traits. DE among species differing in color vision was enriched for several metabolic processes but not neural signaling processes, perhaps suggesting that any expression differences in the V1 influenced by differences in color visual perception are driven more by metabolic differences than by those of synaptic signaling. However, this phenotype–DE comparison shows the strongest influence of phylogeny, and we are not able to parse out if this difference represents that of a divergent color perception or a difference between platyrrhines and cercopithecoids and hominoids.

Furthermore, we chose to investigate the link between group size differences and V1 gene expression due to the hypothesized influences of social behavior on primate brain evolution and the possible link to the required differences in visual perception among group-living primates [19]. While the V1 is not explicitly involved in behavioral processes, there were enrichments for pathways known to be involved in behavior. Primates exhibit an extensive variation in social traits, and a number of genes (and associated pathways) have been hypothesized to be linked to these behaviors, such as those involved in social bonding or empathy (arginine vasopressin receptor 1A (*AVPR1A*), oxytocin receptor (*OXTR*), and dopamine receptor (*DRD4*)) [87]. Oxytocin signaling changes in response to social interactions are well documented [88,89], and variations in the coding sequence for the oxytocin receptor and their associated influence on social behavior have been observed in rhesus macaques [90]. While group size was not enriched for “oxytocin signaling pathway” or “dopaminergic synapse”, the genes *DRD4* and *OXTR* were both differentially expressed in the DE comparison of species differing in group size and alternatively included in the enriched category “neuroactive ligand receptor interaction” (Appendix A). In summary, we show here that genes putatively important in primate social evolution and associated processes display significant DE in the V1 in primates differing in group size. However, as previously mentioned, because there is significantly less variation in expression between cortex regions than between species, future studies comparing multiple brain regions would determine if this were V1 region-specific or a brain region-independent observation. Given the highly organized and specific cytoarchitecture of the V1 [9] and the previously determined influence of structural genes on V1-specifc gene expression in comparison to other regions [46], it is possible that V1 region-specific changes in gene expression are linked to maintaining or fine-tuning this cytoarchitecture. However, studies of V1 function are largely limited to tissue-level functional, mechanistic, and cytoarchitectural investigations [28,91,92] primarily only in rhesus macaques, with very limited understanding of the gene expression changes accompanying an altered system-level function. This, in addition to a lack of focus on the V1 region in leu of other brain regions and a general finding that V1 gene expression is similar to other brain regions except the cerebellum in previous comparative gene expression studies, limits our ability to draw strong conclusions about V1 region-specific gene expression and its link to variations in visually relevant phenotypes.

## 5. Conclusions

Our study investigated the interaction between genotype and phenotype by examining the correlation between gene expression and phenotypic and behavioral traits, including habitat use, color visual system, group size, and diet, in a broad sampling of primate species, including many understudied species (e.g., siamang, squirrel monkey, and spider monkey). We determined that neural and metabolic processes known previously to differ between species in other brain regions also demonstrate interspecies and trait-based differences in the V1. We show that human and chimpanzee are outliers for V1 gene expression, differing significantly more in neuron-specific processes related to synaptic signaling than other species do. Although these species appear to be the most divergent, they do not exhibit any major differences in the visually relevant phenotypes investigated here, for which we were able to determine significant expression differences. Future studies that include other primate taxa could further investigate the link between differences in primate vision evolution and visual cortex expression differences.

As primates exhibit many unique visual system traits compared to other mammals, understanding the genetic basis for primate visual systems in the V1 region would provide valuable insights into the evolutionary trajectory of these traits. Our data indicate that there is also a correlated difference in gene expression in the initial processing center of visual signals in the brain. We also show that humans differ in brain gene expression in the V1 in a manner like other regions. Further investigation of the overlap between DE and signals of selection can provide information about which expression changes are adaptive.

## Figures and Tables

**Figure 2 genes-16-00117-f002:**
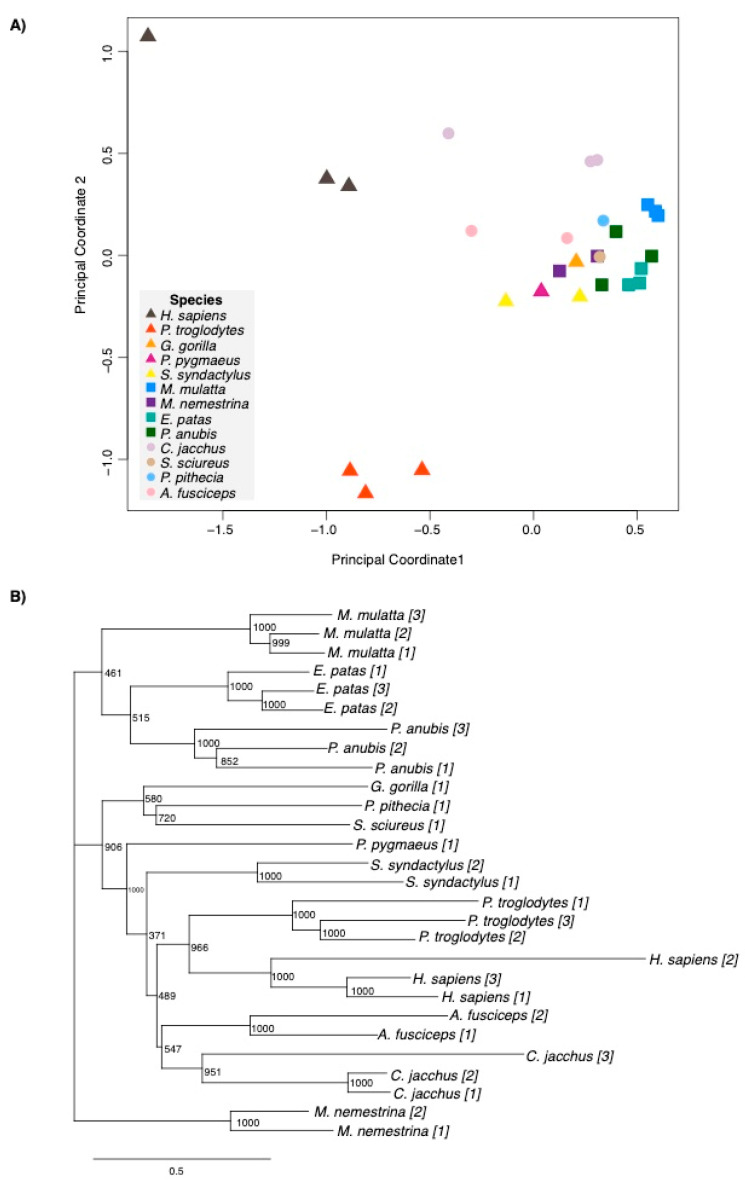
Humans and chimpanzees are the most divergent in V1 gene expression. (**A**) Principal coordinate analyses (PCoAs) of V1 transcriptomes color-coded by species. The shapes of points indicate clade: triangles for hominoids, squares for cercopithecoids, and circles for platyrrhines. (**B**) Hierarchical clustering of V1 transcriptomes of each sample with bootstrap values and the individual sample number in brackets.

**Figure 3 genes-16-00117-f003:**
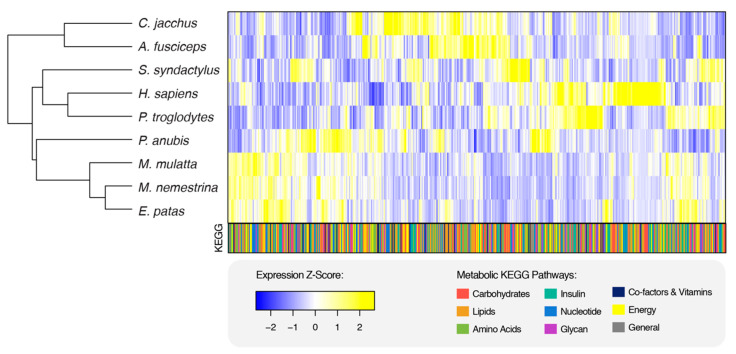
Expression profiles of metabolic genes in primate V1s. Clustering of expression profiles of 1039 metabolic genes in primate V1s. Highly correlated genes (columns) cluster together, and samples (rows) cluster based on the Euclidean distance between expression values. Only species for which there were greater than one sample per species were used. Averages of expression per gene were calculated across replicates per species. The bottom bar represents membership in the color-coded KEGG metabolic pathways for each gene in the heatmap.

**Table 1 genes-16-00117-t001:** Metabolic pathways are the most significantly enriched in interspecies DE comparisons with the exception of human–chimpanzee. A table of the top 10 most significantly enriched KEGG pathways in interspecies DE comparisons between human and other species, abbreviated to show only metabolic (green) and neuron-specific (purple) pathways. “# Genes” refers to the number of genes exhibiting DE in that pathway. *p*-value rank refers to the rank of the enrichment within the top 10 for each comparison. The table is sorted by interspecies DE comparison. Abbreviated from Appendix A.

Interspecies–DE Comparison	KEGG Term	KEGG Parent Category	# Genes	Enrichment *p*-Value	*p*-Value Rank
Human–Chimpanzee	Cholinergic synapse	Neuon-specific:—Synaptic	40	0.00001	1
Human–Chimpanzee	GABAergic synapse	Neuon-specific:—Synaptic	31	0.00011	4
Human–Siamang	Neuroactive ligand receptor interaction	Neuon-specific:—Signaling	75	0.00086	3
Human–Siamang	Metabolic pathways	Metabolism—General	272	0.00730	9
Human–Baboon	Metabolic pathways	Metabolism—General	364	0.00001	1
Human–Baboon	Fatty acid metabolism	Metabolism—Lipids	20	0.00558	7
Human–Rhesus macaque	Steroid biosynthesis	Metabolism—Lipids	11	0.00118	5
Human–Rhesus macaque	Purine metabolism	Metabolism—Nucleotides	55	0.00206	7
Human–Rhesus macaque	Nicotinate and nicotinamide metabolism	Metabolism—Cofactors and Vitamins	13	0.00465	10
Human–Marmoset	Metabolic pathways	Metabolism—General	349	0.00046	3
Human–Marmoset	Nicotinate and nicotinamide metabolism	Metabolism—Cofactors and Vitamins	15	0.00122	6
Human–Spider monkey	Metabolic pathways	Metabolism—General	326	0.00024	1
Human–Spider monkey	Steroid biosynthesis	Metabolism—Lipids	11	0.00142	3
Human–Spider monkey	Purine metabolism	Metabolism—Nucleotides	54	0.00549	7

**Table 2 genes-16-00117-t002:** Proportions of DE genes in primate species V1s vary depending on the compared phenotypes. Pairwise list of phenotype-based DE comparisons. “# Genes” refers to the number of genes exhibiting DE in that pathway. This is a subset of all comparisons made; see Appendix A for the full list of genes exhibiting DE for all comparisons. Abbreviated from Appendix A.

Phenotype–DE Comparison	# Genes	% Genes
Color vision	3173	25.25
Habitat-use	2501	19.91
Group-size	1704	13.56
Folivore–frugivore	534	4.25
Folivore–omnivore	664	5.28
Frugivore–omnivore	991	7.89
Sex	35	0.28

**Table 3 genes-16-00117-t003:** Some KEGG pathway enrichments in phenotypic–DE comparisons are not enriched in more than one phenotype–DE comparison. “Uniquely enriched” KEGG terms are shown for each phenotype–DE comparison investigated. KEGG parent categories do overlap, with the exception of “Cofactors and Vitamins”. “# Genes” refers to the number of genes exhibiting DE in that pathway. Abbreviated from Appendix A. Purple highlights metabolic categories and green highlights neuron-specific pathways.

Phenotype–DE Comparison	KEGG Term	KEGG Parent Category	# Genes	*p*-Value
**Metabolic Pathways**				
Color vision	Biosynthesis of amino acids	Amino Acids	21	0.004
Diet: frugivore–omnivore	Glycine serine and threonine metabolism	Amino Acids	5	0.046
Group size	Histidine metabolism	Amino Acids	6	0.013
Color vision	Glycolysis Gluconeogenesis	Carbohydrate	17	0.030
Diet: frugivore–folivore	Retinol metabolism	Cofactors and Vitamins	5	0.030
Color vision	Sphingolipid metabolism	Lipids	13	0.028
Diet: folivore–omnivore	Steroid hormone biosynthesis	Lipids	6	0.012
Group size	Fatty acid degradation	Lipids	8	0.031
Habitat use	α Linolenic acid metabolism	Lipids	7	0.030
Habitat use	Ether lipid metabolism	Lipids	11	0.020
Habitat use	Glycerophospholipid metabolism	Lipids	20	0.013
Color vision	Pyrimidine metabolism	Nucleotides	26	0.012
Diet: folivore–omnivore	Purine metabolism	Nucleotides	12	0.015
**Neuron-Specific Pathways**				
Diet: frugivore–omnivore	Serotonergic synapse	Synaptic	12	0.010
Diet: frugivore–omnivore	Synaptic vesicle cycle	Synaptic	8	0.012
Group size	GABAergic synapse	Synaptic	14	0.016

## Data Availability

All data are available on the Short Read Archive: https://www.ncbi.nlm.nih.gov/bioproject/?term=PRJNA639850.

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
