# Peer review of "Ecological Trait Differences Are Associated with Gene Expression in the Primary Visual Cortex of Primates"

_genes, 2025, doi:10.3390/genes16020117_

Round 1

Reviewer 1 Report

Comments and Suggestions for Authors

Babbitt and colleagues present an interesting, but preliminary, communication regarding gene expression in the primary visual cortex (V1)of individuals from 13 different species of primates and explore the relationships between differentially expressed transcriptomes and four traits:  namely, type of color vision, habitat, group size and primary diet.  The rationale for this study is very well enunciated and the authors call attention to the paucity of interspecies studies of this type in the literature.  The study included 5 species of hominoids, 4 of cercopithecines and 4 of platyrrhines. 

The study utilizes high level molecular tools and well documented analytical methods though neither the primary nor the secondary data were available for review.

Three major results are presented:  (1) humans and chimpanzees exhibited the greatest degree of variability (differential expression, DE) in V1; (2) across species, variation in metabolic pathways accounted for 23.8% of DE enrichments and neuronal pathways accounted for only 7.6% of DE enrichment and (3) DE in visual cortex was correlated with all 4 of the investigated traits.  With respect to (3), color vision and habitat use (terrestrial vs arboreal) were highly significant while group size and diet were somewhat less prominent.  

While this is unquestionably a worthwhile communication, there are also some limitations that should be more fully discussed and/or highlighted.  While recognizing that sampling a large number of species is difficult, the manuscript would be strengthened by a better description of the condition of the animals from which the samples were drawn, at the time and cause of death, as well as the manner in which the samples were banked.  If all of the samples were derived from captive animals, the extent to which habitat use, group size and diet were important trait features requires further information.  At least as important, and this is highlighted by the results presented, relying on a single sample from a given species may or may not provide an accurate profile of the transcriptome.  This is clear, at least in part, from the variability both within and between phylogenetically closely related species.  In my opinion, the communication would be strengthened by additional discussion of these (and perhaps other) limitations of the study.

Reviewer 2 Report

Comments and Suggestions for Authors

In this interesting study brain tissue samples were collected from 28 individuals from 13 primate species. RNA-sequences of V1 genes were used to estimate differential expression among the species, differing also in their visual phenotypes and broad ecological traits. Results suggested that genes were differentially expressed in relation to the type of colour vision, habitat use, group size and primary diet. Humans and chimpanzees were the most divergent species.

The results of this study are interesting, but preliminary. The sample sizes are too small to obtain even a minimal estimate of intraspecies variability. The absence of estimates of intraspecific variability and the origin of the primates from which the samples were obtained could have introduced biases in the results. The authors have correctly identified some of these potential biases. But the Discussion should probably be more explicit.

In particular:

Introduction

Line 122 - We observed differences in V1 gene expression associated with variation in the visually relevant traits of color vision, habitat use, group size, and diet, but not sex.

Please, specify if the observed differences are statistically significant.

Material and Methods

L. 131 - Frozen brain samples from captive adult primates free of known neurological disorders were obtained from various research institutions and zoos.

The conditions of collection, sampling, and storage of tissues could have influenced the preservation of RNA and therefore the results of the analyses. The authors should discuss these points in more detail and possibly provide indications that all samples were in good preservation conditions. For instance:

L. 150 - RNA integrity varied among our samples due to sampling from deceased primates, but there was no bias for species.

What does it mean? Please, add some more details

Results

Add the number of sequenced samples to the tree in Fig. 1

Discussion

L. 554 – 559 … We found that global gene expression in primate V1 clusters largely by phylogenetic relatedness

Well, but Intraspecific variability – if any – is missing

As correctly noted in Materials & Methods

L. 217-240 - We used species-typical categories for each of the phenotypes … … but not from dynamic experience-dependent differences, which would probably yield even stronger signals of ecological correlations.

Maybe it is better to move this paragraph to Discussion or discuss these issues more deeply in Discussion.

Round 2

Reviewer 1 Report

Comments and Suggestions for Authors

The investigators have replied appropriately to my concerns.

Reviewer 2 Report

Comments and Suggestions for Authors

This ms. has been carefully revised following reviewers' suggestions.